# Greedy Mechanism Based Particle Swarm Optimization for Path Planning Problem of an Unmanned Surface Vehicle

**DOI:** 10.3390/s19214620

**Published:** 2019-10-24

**Authors:** Junfeng Xin, Jiabao Zhong, Shixin Li, Jinlu Sheng, Ying Cui

**Affiliations:** 1College of Electromechanical Engineering, Qingdao University of Science and Technology, Qingdao 266061, China; zhongjbchn@163.com (J.Z.); seawolf_lsx@163.com (S.L.); 2Transport College, Chongqing Jiaotong University, Chongqing 400074, China; seawolf_algorithm@163.com

**Keywords:** particle swarm optimization, greedy mechanism, 2-opt operation, path planning problem, unmanned surface vehicle

## Abstract

Recently, issues of climate change, environment abnormality, individual requirements, and national defense have caused extensive attention to the commercial, scientific, and military development of unmanned surface vehicles (USVs). In order to design high-quality routes for a multi-sensor integrated USV, this work improves the conventional particle swarm optimization algorithm by introducing the greedy mechanism and the 2-opt operation, based on a combination strategy. First, a greedy black box is established for particle initialization, overcoming the randomness of the conventional method and excluding a great number of infeasible solutions. Then the greedy selection strategy and 2-opt operation are adopted together for local searches, to maintain population diversity and eliminate path crossovers. In addition, Monte-Carlo simulations of eight instances are conducted to compare the improved algorithm with other existing algorithms. The computation results indicate that the improved algorithm has the superior performance, with the shortest route and satisfactory robustness, although a fraction of computing efficiency becomes sacrificed. Moreover, the effectiveness and reliability of the improved method is also verified by its multi-sensor-based application to a USV model in real marine environments.

## 1. Introduction

Unmanned Surface vehicles (USVs), also known as autonomous surface crafts, have attracted worldwide attention in commercial, scientific and military fields recently. USVs have the advantages of low operation and maintenance costs, reduced casualty risk, and good maneuverability and deployability for different operating conditions [1]. With the aid of effective and reliable navigation equipment, such as global positioning systems (GPS), wireless communication units, and various types of sensors, USVs can be employed cost-efficiently for a variety of applications, including underwater surveying [2], pollutant tracking [3], acoustic navigation [4], marine rescue [5], and obstacle detection [6]. Although rapid progress of corresponding techniques for USV systems have been achieved in recent times, it is still challenging in promoting the level of USV autonomy when faced with complex or hazardous environments. Key issues, including hull hydrodynamics, communication techniques, and navigation, guidance and control (NGC) strategies, require further developments.

As a vital component of USV guidance systems, path planning is of critical importance in designing and updating feasible and optimal trajectories for the control system on basis of navigation information, mission demands, and environmental conditions. Its effectiveness not only determines the autonomy of unmanned vehicles, but also affects the reliability and efficiency of mission execution. From the literature, plenty of intelligent techniques have been employed in the USV path planner [7]. Meanwhile, it is a research hot pot to propose an algorithm having fast convergence speed, admirable robustness, low computation consumption, and satisfactory route planning.

The USV path planning problem is normally formulated as a travelling salesman problem (TSP), which is a typical combinatorial optimization problem. The specific description is to find a shortest loop that passes through all target cities without repetition. However, when a USV performs multi-objective tasks in a complex marine environment, the number of possible paths increases exponentially with the increase of target points’ number, resulting in the so-called “exponential explosion”. In this circumstance, traditional algorithms, such as the exhaustive method and branch-and-bound algorithm, are unable to find the optimal solution within reasonable time cost. Hence, it is vital to develop efficient and feasible heuristic algorithms instead of traditional methods to solve this kind of problem, although the optimality, accuracy, and completeness becomes sacrificed for running speed to some degree. Commonly used heuristic algorithms include elephant search algorithm (ESA) [8], genetic algorithm (GA) [9], particle swam optimization (PSO) [10], firefly algorithm (FA) [11], grey wolf optimizer (GWO) [12], and so on.

However, the conventional type of each method has its inherent limitation. For instance, it is known that the ESA provides a random and improper replacement for an elephant with the worst fitness, which is likely to result in the deterioration of search space. The conventional GA has inherent issues of slow convergence speed, poor capability of local search and easy occurrence of premature convergence. In other words, the loss of population diversity is likely to appear during the implementation of GA, making individual genes tend to be same and terminating the search at local optimum. Additionally, the use of a real-number code also generates repetitive genes during crossover operations, which affects convergence speed greatly. For the PSO, the lack of a balanced mechanism would lead to the loss of population diversity and finally being trapped into the local optimum. In addition, the FA only considers the distance and maximum attractive force between two fireflies, and has intrinsic shortcomings of poor search capability during early iterations and severe oscillations around the optimal solution during later iterations. In terms of the GWO, the search mechanism of judging distance from prey would lead to a slow convergence speed for later iterations. Nowadays, it has become a tendency to combine two or more algorithms, making full use of their respective advantages to enhance the algorithm effectiveness. 

Inspired by the combination strategy, this work optimizes the conventional PSO method by introducing a greedy mechanism to generate partial particles and 2-opt operation to eliminate path-crossing phenomena. Comparative study with some state-of-the-art algorithms is conducted to verify the effectiveness and reliability of the improved method in terms of solution quality, algorithm robustness, and computing efficiency. Then the improved algorithm is adopted to generate a feasible closed trajectory for a self-developed USV to follow in real marine environments.

It is worth noting that this work has certain connections and essential differences with our two recently published works [13,14]. In the first work, we optimized GA by multi-domain inversion for the purpose of increasing the number of offsprings. The inspiration comes from the biological theory foundation, where the number of offsprings needs to be larger than the number of parents so as to prevent species extinction and maintain species diversity in the process of biological evolution. In the other work, we optimized PSO by iteratively adjusting the control parameters of the PSO algorithm, such as inertia weight and acceleration coefficients, and introducing merit-based selection and uncertain mutation. The optimization is from the point of view of global searching. However, in this work, we start form the inspiration of the combination strategy of several intelligent algorithms and introduce the greedy mechanism and 2-opt operation in order to optimize the local search of PSO.

The main contributions of this work are as follows: (1) a greedy black box is established to generate the initial swarm of particles which avoids the randomness of the traditional method; (2) the strategy of greedy selection guarantees particles to move towards a higher fitness level and keeps the swarm diversity; (3) the 2-opt operation performs effectively in maintaining locally optimal fragments of relatively inferior particles and eliminate path crossovers; (4) the improved algorithm has been successfully applied to the path planning subsystem of a USV model with the aid of multi-sensors.

The rest of the paper is organized as follows. A brief literature review is presented in Section 2 on the research status of the GSM and PSO methods. The conventional and improved PSO algorithms are introduced concisely in Section 3. Results of Monte-Carlo simulations and sea trials are given in Section 4. Conclusions are drawn and future researching interests are suggested in Section 5.

## 2. Literature Review

### 2.1. Greedy Algorithm

The principle of a greedy algorithm is generally depicted as making a choice which seems locally optimal at present and intending to find a satisfactory solution in a reasonable amount of time. The choice may refer to previous decisions, but never depends on future choices or the other choices inherent to the subproblem. In other words, the greedy algorithm is short sighted and never reconsiders its decisions. This is totally different from dynamic programming which would reconsider the previous decisions, and is guaranteed to find the optimal solution after exhaustive operations. In addition, the greedy algorithm mostly fails to achieve the global optimum. Nevertheless, it is still widely used as an auxiliary strategy or to give good approximations of the optimum for some problems with time constraints. Meanwhile, the greedy algorithm performs effectively for problems of optimal substructure whose globally optimal solution contains locally optimal solutions to subproblems. 

For solving the TSP, the greedy algorithm is implemented by choosing the nearest unvisited city as the salesman’s next target until a completely closed route is generated. Since achieving the optimum for the TSP is typically time-consuming, the greedy algorithm has been integrated into various algorithms to accelerate algorithm convergence and to enhance the search efficiency in recent times. Pan et al. proposed a hybrid algorithm which employed immune algorithm for global searching, a greedy algorithm for population initialization, and a delete-cross operator for eliminating path crossovers. Results verified that this hybrid method enhanced the reliability, global convergence speed, and search ability of immune algorithm [15]. In order to reduce computational time with the premise of guaranteeing solution quality, Basu et al. adopted a randomized greedy contract method as the preprocessing step for image sparse in advance of using tabu search to solve the asymmetric TSP. Comparative study with other success heuristics methods found that the optimization exhibited the potential to shorten the time cost by 1%–5% [16]. Mestria integrated the greedy randomized adaptive search procedure (GRASP), iterated local search, and variable neighborhood descent, proposing a hybrid method for clustered TSP. The hybrid heuristic outperformed several existing methods in terms of optimal solution and reasonable computational time for medium and large size instances [17].

### 2.2. Particle Swarm Optimization

PSO was proposed by Eberhart and Kennedy in 1995, inspired by the bird flight model [18]. Aiming at obtaining the global optimal solution, a flock of birds was abstracted to a swarm of random particles without quality and volume parameters to prevent the collision phenomenon of real birds. 

Each particle moves around in the search space under the dynamic guidance of its own flight state, its own experience and the swarm’s experience. By use of a fitness function for solution evaluation, the swarm is expected to advance towards the most satisfactory solution. Moreover, it is prevalently known that the PSO has advantages of fast convergence, simple parameter settings, and easy implementation, and has been widely applied to a large number of discrete and continuous optimization problems, such as the TSP [10], the molecular docking problem [19], electrochemical machining [20], image recognition [21], semi desirable facility location problem [22], sheet metal forming [23], and so on.

For the purpose of enhancing the PSO effectiveness in solving the TSP, plenty of valuable research has been conducted on the combination of two or three heuristic algorithms. Shuang et al. modified the ant colony optimization (ACO) algorithm by introducing the PSO to expand the search space and to use the swarm experience for convergence acceleration [24]. The similar strategy of introducing PSO into ACO was also verified by Mahi et al. in order to optimize the city selection parameters [25]. Another improved PSO, proposed by Zhang et al., adopted a priority coding method to code the solution, set the velocity range dynamically and to eliminate the side effect of discrete search-space, which applied the k-centers method to avoid being trapped into the local optimum. This combinational algorithm had a good performance in the reservation of swarm diversity [26]. In the work by Feng et al., the large-scale search space was first divided into subspace by the adaptive fuzzy C-mean algorithm. Thereafter, transform-based PSO and simulated annealing methods were used together for finding the locally optimal solution. The complete route was finally rebuilt by the max-min merging algorithm [27]. Admirable results were also achieved by the trials of combining PSO with the genetic algorithm and artificial fish swarm method, respectively [28,29].

### 2.3. Combinations of Greedy Algorithm with PSO

As mentioned above, on the one hand, the PSO has inherent limitations whereby the algorithm is easyily trapped into the local optimum, and is not that effective at solving discrete problems. On the other hand, the greedy algorithm has a relatively simpler principle, easier implementation, and higher efficiency, though it is not guaranteed to produce a globally optimal solution. Therefore, based on the idea of complementary advantages, it seems entirely feasible to combine the two algorithms for various optimization problems. 

Bajwa et al. developed a hybrid method for jobs scheduling on a single machine in a group technology system, aiming at minimizing tardy jobs without preemption. The GRASP was adopted to enhance search efficiency, and the resulting suitable solutions were further employed as particles by the PSO algorithm to achieve local and global solutions. This hybrid heuristic was proved to be effective for 63% of problem instances [30]. In order to solve vehicle routing problems with time windows, Marinakis et al. presented a new variant of PSO based on three strategies: GRASP for solution initialization, adaptive combinatorial neighborhood topology for particles movement, and adaptive calculation of parameters. By comparing with other effective algorithms from the literature, the proposed method was proved to be more flexible, faster, and more effective [31]. Furthermore, an improved PSO was proposed by Pathak et al. for minimum zone form error evaluation. Greedy selection procedure was used to guide the update of newly-generated solutions at each iteration. Results validated the effectiveness of the new algorithm, especially in improving exploitation capability [32]. Another successful application of the greedy algorithm to the PSO method could be found in the work by Ali-Askari et al., aiming at addressing a location-allocation problem for a capacitated bike sharing system with demand uncertainties [33].

## 3. Algorithms

### 3.1. Particle Swarm Optimization

The conventional PSO scatters *N* particles randomly within an *S*-dimensional search space at the beginning of evolutionary process. For the *i*-th particle, its position and velocity can be represented by two vectors *X_i_*= (*x_i_*_1_*, x_i_*_2_*, ..., x_iS_*)*^T^* and *V_i_* = (*v_i_*_1_*, v_i_*_2_*, ..., v_iS_*)*^T^*, respectively. The fitness function is defined as 1/*D_i_* (*D_i_* stands for the route length), and the individual best-known position (*P_is_*) and swarm best-known position (*P_gs_*) are updated during every iteration. At the same time, the velocity and position of each particle are updated as [34,35].
(1)$$v_{is}^{m+1}=wv_{is}^{m}+c_{1}r_{1}^{m}\left(P_{is}^{m}-x_{is}^{m}\right)+c_{2}r_{2}^{m}\left(P_{gs}^{m}-x_{is}^{m}\right),$$
(2)$$x_{is}^{m+1}=x_{is}^{m}+v_{is}^{m+1},$$
where *m* stands for the current number of iteration and *s* is the *s*-th dimension, respectively. *r*_1_ and *r*_2_ are random numbers which are iteratively-updated and uniformly distributed between 0 and 1. *c*_1_ and *c*_2_ are the personal cognition coefficient and social cognition coefficient, and are usually suggested with 2. *w* is the inertia weight. The pseudocode of conventional PSO is described in Algorithm 1.

It should be noted that PSO control parameters determine the balance degree between exploration (searching a broader space) and exploitation (moving to the local optimum), which has a significant impact on the algorithm performance. For instance, acceleration coefficients *c*_1_ and *c*_2_ play an important role in balancing the effects of personal cognition and social cognition on guiding particles towards target optimal solution. The stochastic characteristics of *r*_1_ and *r*_2_ can maintain population diversity and avoid premature convergence to some degree. Nonetheless, the choice or dynamic adjudgment of PSO parameters are not the focus of this work.


**Algorithm 1: Conventional Particle Swarm Optimization for TSP**


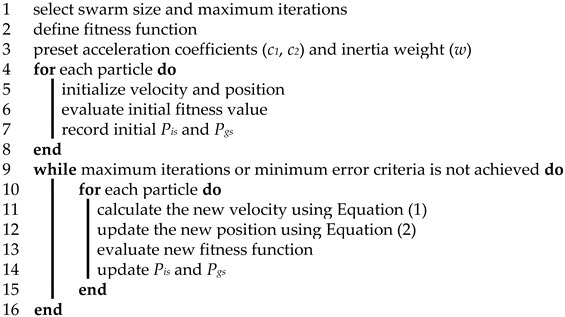



### 3.2. Improved Particle Swarm Optimization

It is known that the conventional PSO generates an initial swarm of particles in a random way, and that this results in some infeasible solutions and consequently restricts convergence speed and search efficiency. Hence, it is expected to enhance the algorithm’s effectiveness by integrating two local strategies, greedy mechanism and 2-opt operation, into the PSO. In our new algorithm, a greedy black box is utilized for the stage of particle initialization and particle generation at each iteration. However, since the greedy mechanism only considers current circumstances for local searches, it has inherent shortcomings that path crossovers are easy to arise. Hence, it is subsequently necessary to adopt 2-opt operation in order to enhance the probability of satisfactory combinations between locally optimal fragments, and to eliminate path crossovers. 

#### 3.2.1. Greedy Black Box for Particles Initialization

A greedy black box is established based on the greedy mechanism. It has the function to build a locally optimal solution once the departure city is determined. For the TSP with *N* cities, we can discover *C* cities which are nearest to the *i*-th city through distance ranging. Thus, a matrix *A_N_*_× *C*_ is defined with the size of *N* × *C*. The element *A_ij_* represents the code of city which is the *j*-th nearest city to the *i*-th city. The line *A*[*i*] is the code of *C* cities which are nearest to the *i-*th city. 

In this work, the value of *C* is set as 3. By using burma14 from the library of sample instances for traveling salesman problem (TSPLIB) as an example, the mechanism of the greedy black box can be described as follows. If the salesman starts from the third city, the code of 14 is selected from line *A*[3] as the next city to visit. Then from line *A*[14], the twelfth city is the nearest one and should be chosen as the next target city. The next target city is the sixth city as shown in line *A*[12]. By analogy, the following cities would be selected one by one to form the travelling route. During this process, repeatability inspection is conducted to guarantee that each city would be visited only once. Usually, three columns could satisfy the requirement of generating a new solution. However, if repetition is inevitable, the next target city would be selected randomly for a repeatability inspection until a closed route is formed. Based on this principle, the initial travelling route could be represented by a string of city code *L*= {3, 14, 12, 6, 7, 13, 8, 1, 11, 9, 10, 2, 5, 4}, and is illustrated in Figure 1.

The matrix *A*_14 × 3_ is given as:(3)$$A_{14\times 3}=\left(\begin{array}{ccc} 8 & 11 & 9\\ 1 & 8 & 11\\ 14 & 4 & 12\\ 12 & 3 & 14\\ 6 & 12 & 7\\ 12 & 7 & 14\\ 13 & 12 & 6\\ 1 & 11 & 9\\ 11 & 1 & 8\\ 9 & 11 & 1\\ 9 & 8 & 1\\ 6 & 14 & 7\\ 7 & 8 & 12\\ 12 & 3 & 6 \end{array}\right)$$

On the one hand, the utilization of the greedy black box excludes a great number of infeasible solutions resulted from the randomness of the traditional method, and creates an initial swarm of particles with a relatively higher quality. On the other hand, the strategy of the greedy black box makes the newly planned route consist of several locally optimal fragments, with advantages of short distance, low order, and high adaptability. Hence, it seems feasible and reasonable to adopt the greedy black box for the particles initialization.

#### 3.2.2. Greedy Selection Strategy and 2-opt Operation for Local Search

Local search is used further to improve the quality of generated solutions. After the update of particles’ velocity and position during an iteration, all the particles are scrambled and divided into sub groups of four. The “out of order” execution avoids repeating the later replacement of a same particle as much as possible. Meanwhile, the operation of grouping helps establish a communication platform within a subgroup or between arbitrary two subgroups. Then two particles with lower fitness values in each sub group are chosen. Their first cities are utilized by the greedy black box to generate two new particles. If the new generated particles have higher fitness values, they will replace the old ones; otherwise, the replacement will not be conducted. This operation results in the survival of the fittest to some degree and prevents the swarm of particles being generated entirely by the greedy black box, and is also called greedy selection strategy. Furthermore, since the particle generation by greedy black box only considers the locally optimal code, selection and non-repeatability, certain route crossovers are likely to occur, for instance, a crossover around the sixth city, as shown in Figure 1. Therefore, 2-opt operation is employed in each iteration in order to eliminate the path-crossing phenomenon. Our expectation is to maintain solutions having locally optimal fragments, and to enhance the probability of satisfactory combinations between these fragments.

The 2-opt operation was first proposed by Croes in 1958 [36]. Its mechanism is illustrated in Figure 2. Two edges are randomly deleted, dividing the closed route into two parts. Then, ends of the two parts are reconnected in the other appropriate way, creating a new solution route. In other words, from the point of view of path encoding, two non-adjacent city nodes are selected randomly. The route fragment between them will be integrally inversed and connected back to the original encoding string, reproducing a new solution. This procedure will be repeated until the shortest route is found.

Theoretically, on the one hand, the 2-opt operation ensures that the newly generated solution would preserve the excellent fragment of encoding strings from the old one. On the other hand, the greedy selection strategy guarantees the unidirectionality of optimizing the solution quality, namely, that particles keep moving towards a higher-fitness level.

#### 3.2.3. Procedures of Improved PSO

The pseudocode of improved PSO is shown in Algorithm 2. First, the greedy black box is built to generate the initial swarm of particles which consists of several locally optimal solutions. During each iteration, the fitness of each particle would be calculated to update the individual and best social solutions. Then, the velocity and position of each particle are updated according to Equations (1) and (2). Moreover, the greedy selection strategy is applied along with the 2-opt operation for the local search. The algorithm will be terminated when the maximum number of iterations has been reached or the adequately short route is found.


**Algorithm 2: Improved Particle Swarm Optimization for TSP**


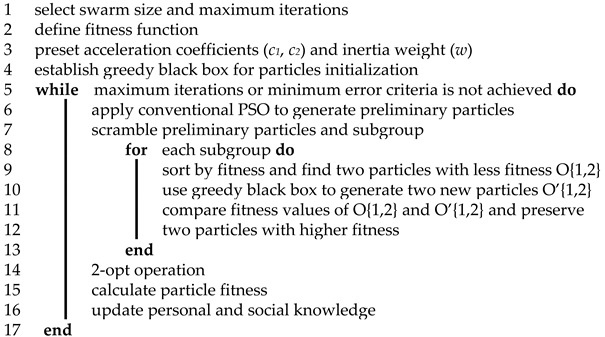



The advantages of improved PSO are evident. First, the particles initialization by the greedy black box overcomes the randomness of traditional methods, and excludes several infeasible solutions at the beginning of optimization. Moreover, in order to keep the population diversity and maintain the locally optimal fragments of relatively inferior particles, four steps are conducted: disordering particles, dividing them into sub groups, sorting the fitness, and generating particles for the next iteration by greedy selection strategy. In addition, the application of the 2-opt operation remedies the defect that the greedy black box only focuses on the locally optimum, and enhances the probability of satisfactory combinations between excellent fragments. The improved method is expected to enhance the probability of obtaining the optimal solution and to improve the solution quality at the same time.

However, the improved algorithm has certain disadvantages from theoretical perspective. Once the code of departing city is determined, the particle generated by the greedy black box is unique. It is an inherent shortcoming since the greedy mechanism only focuses on locally optima, rather than consider its decisions from a global respective. Furthermore, the maximum number of iterations could not be decided exactly when more planned points are considered. In order to guarantee the successful execution of 2-opt for completely eliminating path crossovers, we usually adopt an obviously excessive number base on experience. Meanwhile, we obtained a preliminary relationship between the number of planned points and the reference maximum number of iterations through early tests, as shown in Figure 3.

## 4. Results and Discussions

In this section, the computational results are presented in detail. First, a quantitative comparison between the improved PSO and three existing algorithms is depicted in Section 4.1. Then, in Section 4.2, the improved PSO is applied to design feasible paths for a self-developed unmanned surface vehicle (USV) in real marine environments. Note that all algorithms are implemented in MATLAB^®^ using a i7-7700hq laptop (brand: ASUS, model: FX53VD) with 2.80 GHz and 8.0 GB RAM.

### 4.1. Comparison between Improved PSO with Existing Algorithms

Two other existing algorithms, the conventional genetic algorithm (CGA) and ant colony optimization (ACO) algorithm, are selected to compare with conventional PSO (CPSO) and improved PSO (IPSO) in this section. Eight instances from TSPLIB are used: eil51, rat99, kroa100, lin105, ch150, kroa200, tsp225, and lin318. For the purpose of eliminating algorithm randomness in MATLAB^®^ operation environment, one hundred Monte-Carlo simulations are performed to obtain the data set of the optimal route distance (*D*) for each instance and each algorithm. Furthermore, the control parameters for each algorithm are shown in Table 1 to facilitate the replicability of this work. It is worth highlighting that the maximum number of iterations (*MAX*) depends on the number of planned points. It is determined through preliminary tests, pursuing the balance between complete convergence and economical time consumption. Corresponding with the eight instances, the *MAX*s are 300, 400, 500, 600, 700, 800, 900, and 1000, respectively.

Figure 4 presents the comparative results in the form of box-and-whisker plots. The legend refers to the explanation by Spear [37]. In every plot, a blue range bar is drawn to represent the interquartile range of the data set. It could reflect the degree of data dispersion, or the algorithm robustness to some extent. Moreover, a red line and a plus symbol are drawn in the bar to identify the median and average values of data set. There are also two whiskers extending on both sides of the bar, whose ends stand for the best and worst values, respectively. Results of ACO and IPSO is enlarged at the top right due to their minor difference. Furthermore, detailed information is provided in Table 2, including the known optimal solution (KOS) form of, average value (AVG), standard deviation (SD), relatively percentage error (RPE) and critical iteration number (*M_cri_*). It should be noted that the SD is calculated to show how far all the data points (*D_k_*) depart from the average which is the quantification of algorithm robustness. In addition, the RPE is defined to reflect the gap between the average solution with the KOS from TSPLIB. The best values of the AVG among the four algorithms for the eight instances are given in bold and highlighted in grey. The SD and RPE are defined as:(4)$$SD=\sqrt{\frac{1}{100}\sum_{k=1}^{100}{\left(D_{k}-AVG\right)}^{2}},$$
(5)$$\mathrm{RPE}\left(\%\right)=\frac{\left(\mathrm{AVG}-\mathrm{KOS}\right)}{\mathrm{KOS}}\times 100$$

In general, the CPSO and the CGA obtain worse solutions with larger average values and larger degrees of data dispersion for all the eight instances. There is a vast promotion room for their effectiveness. By contrast, the ACO and the IPSO show similar and satisfactory performance especially when more planned points are considered. By observing enlarged images, the IPSO could find a shorter route despite a relatively worse robustness. For the instance of lin105, the CPSO gives an exaggerated solution with the AVG of 60625 m and the SD of 2978 m. However, with the aid of the greedy mechanism and 2-opt, the IPSO effectively reduces the AVG by 74.9%, and the SD by 89.3%. For tsp225, the average optimal route distance of the ACO and IPSO are 4300 m and 4188 m, respectively. The SD value of the IPSO is only 0.6% larger than that of the ACO. In addition, it is found that the difference between the AVG of the IPSO and the KOS from TSPLIB is below 10% for all the instances.

Moreover, Figure 5 shows the evolution history of the optimal path distance (*D*) against iteration (*m*) for each algorithm. Overall, the curve of each algorithm has similar development tendency. An obvious decrease in *D* occurs with the increase of iterations until a critical iteration number (*M_cri_*) after which the solution achieves convergence. In this work, the criterion of *M_cri_* is utilized to evaluate the convergence speed or computing efficiency of each algorithm. It is found that the CPSO converges rapidly and terminates at a smallest *M_cri_*. Obviously, it results in being trapped into the local optimum with the longest path distance. The CGA has incredibly slow convergence speed for all instances when compared with the other three algorithms. This behavior helps to find a relatively better solution than the CPSO. Meanwhile, the IPSO and ACO are neck to neck in convergence speed and optimal route distance after convergence. For the instance of kroa100, 446 iterations are required for the CGA to achieve the convergence, seven times more than that for the CPSO. As a result, the CGA could reduce the average optimal route distance by about 42.9%. Moreover, the critical iteration numbers for the IPSO and CPSO are 113 and 62, respectively. The 82% increase of *M_cri_* makes the AVG value reduce from 85243.19 to 22981.74 m. It seems necessary to slow down the convergence speed appropriately in exchange for a high-quality solution.

Since the CPSO and CGA would present chaotic routes for the eight TSPLIB instances, only the best trajectories generated by the ACO and IPSO are shown in Figure 6 and Figure 7, respectively. It can be visualized that the routes of ACO have different levels of path-crossing phenomenon, especially when considering more planned points. However, no path-crossing exists in the routes of the IPSO. It is the application of the 2-opt operation and the greedy selection strategy that helps avoid the intersection of route effectively and reduce the path complexity. This is also the reason why a longer path is obtained by the ACO than the IPSO under the same condition. The results indicate that it is crucial to retard the progress of convergence in order to improve the solution quality and to avoid being trapped into local optima.

### 4.2. Multi-Sensor-Based Application to USV Path Planning

The USV model is self-designed and constructed by Qingdao Municipal Engineering Laboratory of Intelligent Yacht Manufacturing and Development Technology, Qingdao University of Science and Technology (QUST). In the design stage, the USV is expected to be used in fields of military development, scientific research, and environmental protection to execute tasks, such as autonomous cruise, water quality sampling, water quality monitoring, and garbage collection. It is of crucial importance to design the feasible and shortest trajectories covering all preset points for the purpose of reducing costs and enhancing efficiency. Generally, path planning can be simplified into the TSP if the prior environmental information is inaccessible and the collision-free restriction is not taken in to account. Hence, in this section, the proposed IPSO is applied to the navigation, guidance and control (NGC) system of a self-developed unmanned surface vehicle, designing feasible trajectories to follow under real sea conditions. Note that the factors of wind, current, waves, and obstacles are not considered in this preliminary work.

The USV model is 1.8 m long and 0.9 m wide, and consists of five underwater bodies. Figure 8 gives a schematic diagram of the USV NGC system. It involves three module subsystems: the navigation data processing subsystem, path planning subsystem, and autopilot subsystem. When the planned points are chosen in a personal computer, the improved PSO algorithm is adopted in the path planning subsystem, generating feasible trajectories for the USV to follow. The planned trajectories will be transferred to the autopilot subsystem, together with navigation information of the direction of the bow and the location data of the USV gathered by multi-sensors such as the electronic compass and GPS. Then a closed-loop controller is adopted to determine the heading and speed of the vehicle until the next target point is reached. More detailed information is provided in our previous published works.

The test site is located at the Qingdao Olympic Sailing Center in Fushan Bay. Due to the limited space, three testing conditions with three numbers of planned points are considered: *Q* = 30, *Q* = 40, and *Q* = 50. The location coordinates of all planned points refer to Table A1, Table A2 and Table A3. The swarm size is set as 500 and the maximum number of iterations is 300.

Optimal trajectories generated by the ACO and IPSO are presented by solid lines in Figure 9 with detailed information listed in Table 3. Certain self-crossing phenomena occur in the planned paths of the ACO for *Q* = 40 and 50, as marked in blue circles. Whereas, the IPSO generates feasible routes with satisfactory lengths and no path-crossing. Since the number of planned points is not beyond 100, the IPSO performs better than the ACO in both the critical iteration number and optimal path distance. When the number is 30, the two algorithms provide the same route with the length of 1100 m. For *Q* = 40, the IPSO converges at *M_cri_*= 33 with the optimal path distance of 1200 m, while the ACO terminates at a larger iteration of 59 with a longer distance of 1214 m. By comparison, the route designed by the IPSO is delivered to the USV model to follow under real sea conditions. The actual paths for the three sets of planned points are shown using the red dashed lines. It could be observed that the real path is smoother than the planned path. The path between any two planned points deviates from the planned trajectory to some degree due to the impact of wind, waves, and currents. However, the successful application of heading correction to the autopilot subsystem would finally guide the USV model to return to the planned points. The feather of this site-specific cruise would play an essential role in the fields of ocean ranching and multi-point water quality monitoring and sampling in vast water.

## 5. Conclusions

When the conventional particle swarm optimization algorithm is applied to solve the TSP or path planning problem for a USV, the population is normally initialized in a random way, which would generate a great number of infeasible solutions and restrict computing efficiency. Hence, this work introduces the greedy mechanism and 2-opt operation inspired by a combination strategy. The purpose is to optimize the conventional algorithm, enhance the probability of obtaining the optimal solution and improve solution quality at the same time. Monte-Carlo simulations for eight TSPLIB instances and application tests to a self-developed USV are conducted to verify the effectiveness and reliability of the improved method. Results are concluded in the following.

Particle initialization by the greedy black box overcomes the randomness of conventional method and excludes several infeasible solutions; this helps to enhance optimization efficiency.The greedy selection strategy guarantees that particles would keep moving towards a higher fitness level, and helps to maintain population diversity to some degree. Meanwhile, the 2-opt operation is constructive to preserve excellent fragments of encoding strings from old solutions, and remove path crossovers effectively.Compared with conventional algorithms, the improved algorithm could enhance robustness greatly and decrease the optimal route length drastically, although a fraction of computing speed is sacrificed for solution quality.The improved algorithm could be successfully applied to the path planning subsystem of the USV model with the aid of multi-sensors. Although actual paths in real sea are not completely consistent with planned paths due to the impact of wind, waves, and currents, the application of heading correction to the autopilot subsystem would finally guide the USV model to return to specified points.

It is noted that the introduction of 2-opt operation adds certain complexities to the algorithm. Hence, more efforts would be made to further reduce the time consumption in the future.

## Figures and Tables

**Figure 1 sensors-19-04620-f001:**
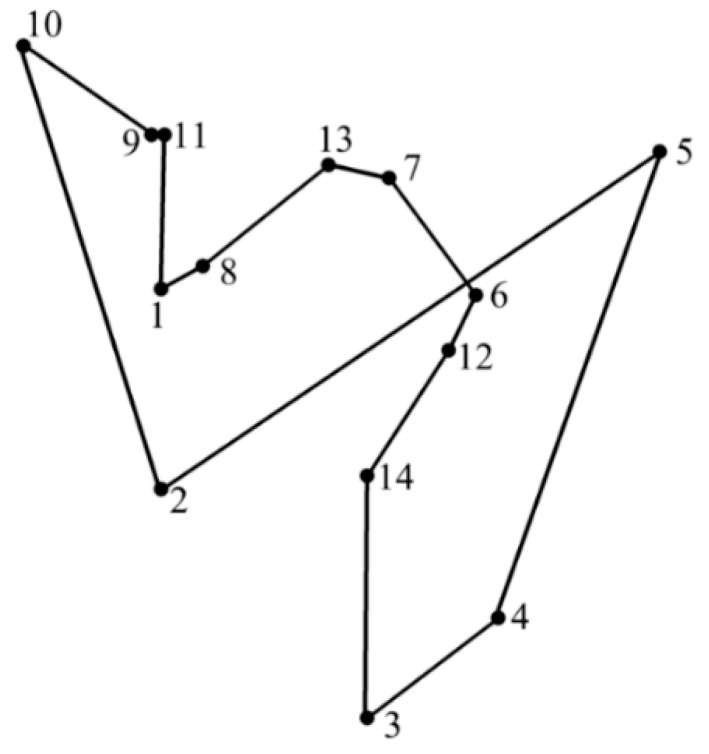
An initial solution of burma14 generated by the greedy black box.

**Figure 2 sensors-19-04620-f002:**
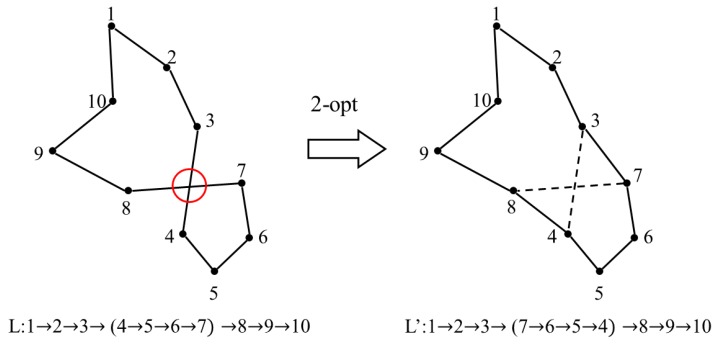
2-opt operation.

**Figure 3 sensors-19-04620-f003:**
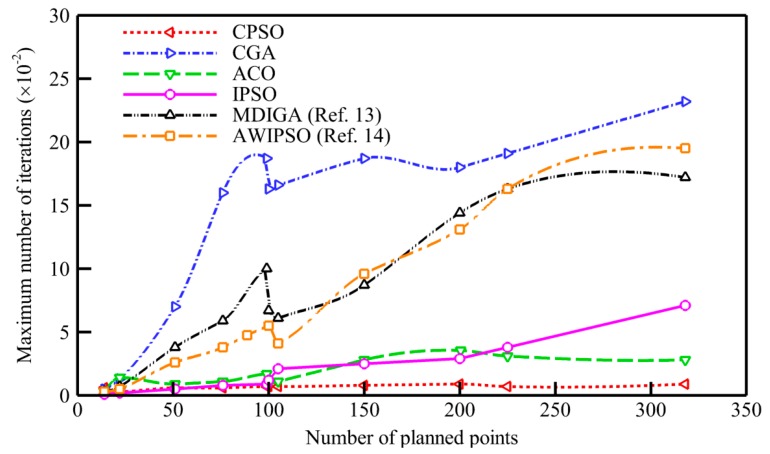
Reference maximum number of iterations for various numbers of planned points.

**Figure 4 sensors-19-04620-f004:**
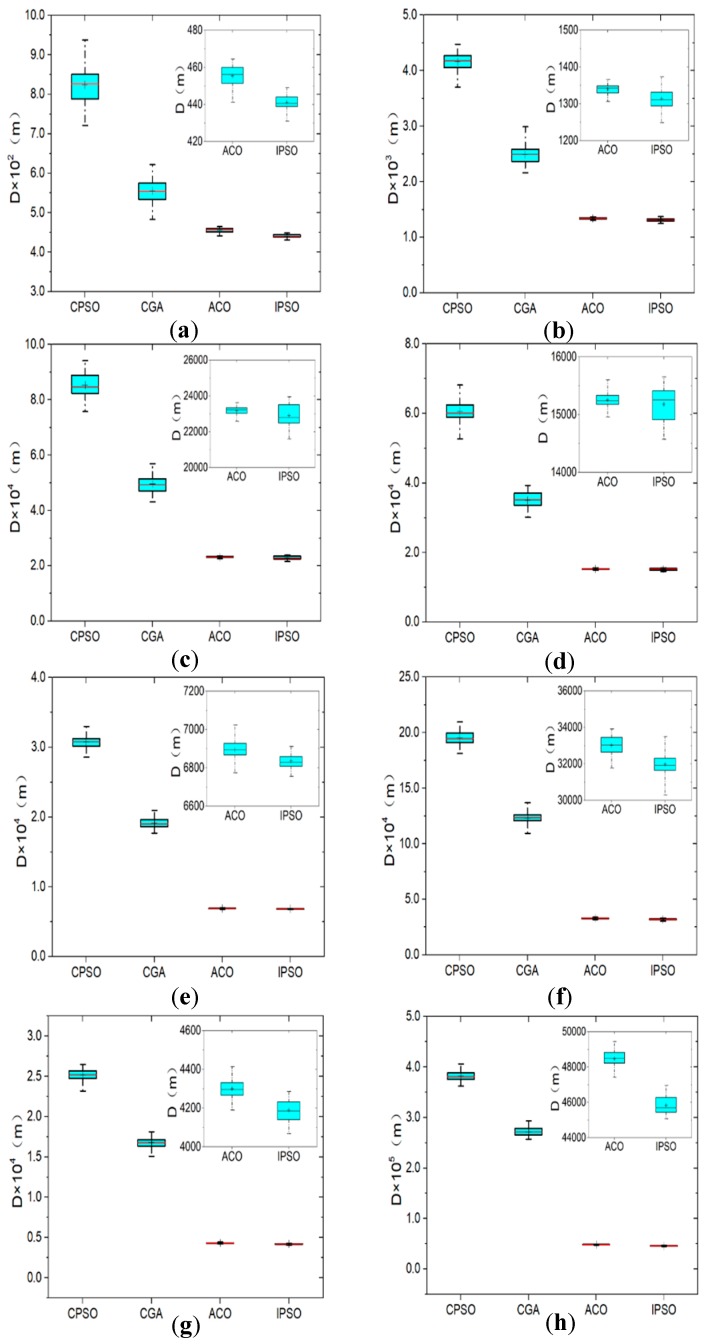
Solution distribution of each algorithm for eight TSPLIB examples: (**a**) eil51; (**b**) rat99; (**c**) kroa100; (**d**) lin105; (**e**) ch150; (**f**) kroa200; (**g**) tsp225; (**h**) lin318.

**Figure 5 sensors-19-04620-f005:**
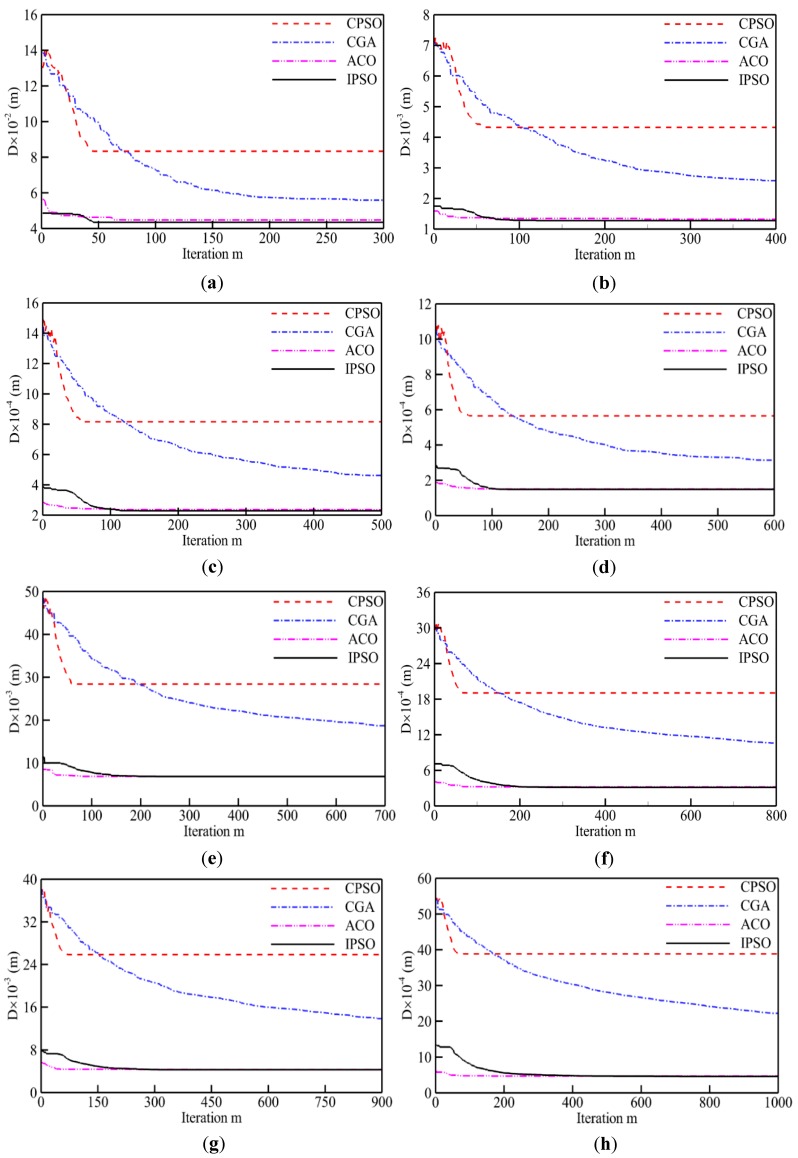
Evolution curves of optimal path distance against iterations for eight TSPLIB examples: (**a**) eil51; (**b**) rat99; (**c**) kroa100; (**d**) lin105; (**e**) ch150; (**f**) kroa200; (**g**) tsp225; (**h**) lin318.

**Figure 6 sensors-19-04620-f006:**
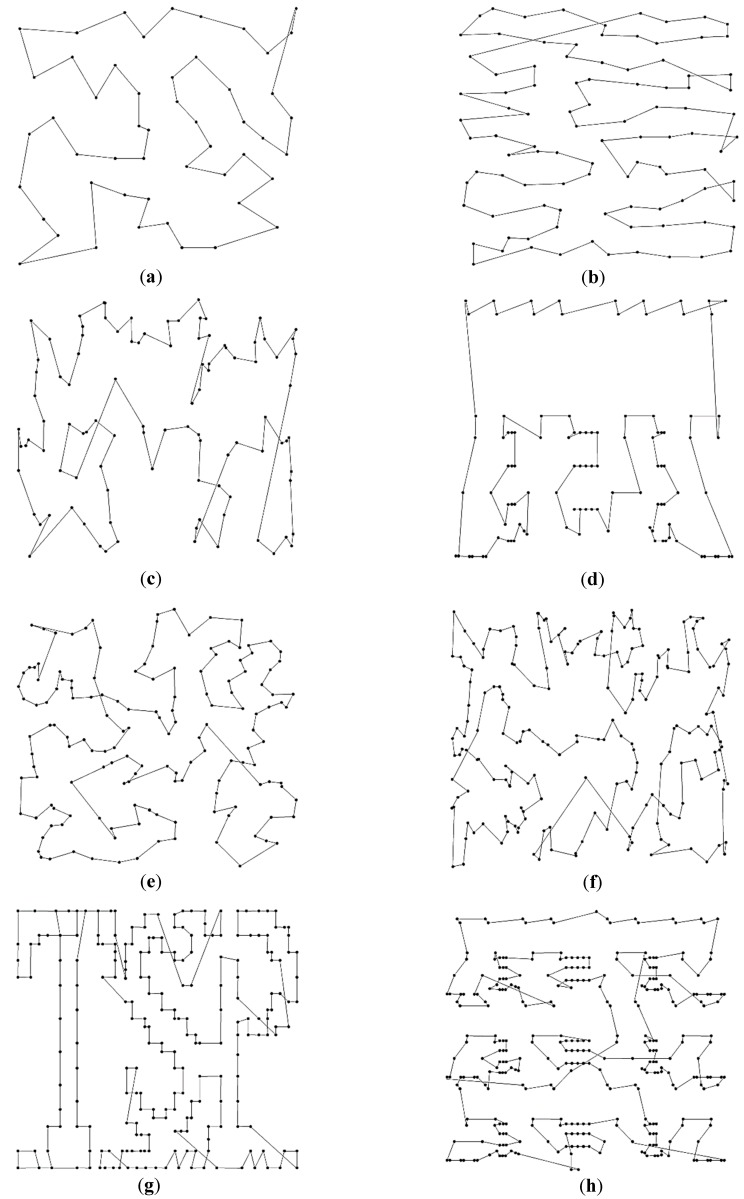
Best route for eight TSPLIB examples using the ACO: (**a**) eil51; (**b**) rat99; (**c**) kroa100; (**d**) lin105; (**e**) ch150; (**f**) kroa200; (**g**) tsp225; (**h**) lin318.

**Figure 7 sensors-19-04620-f007:**
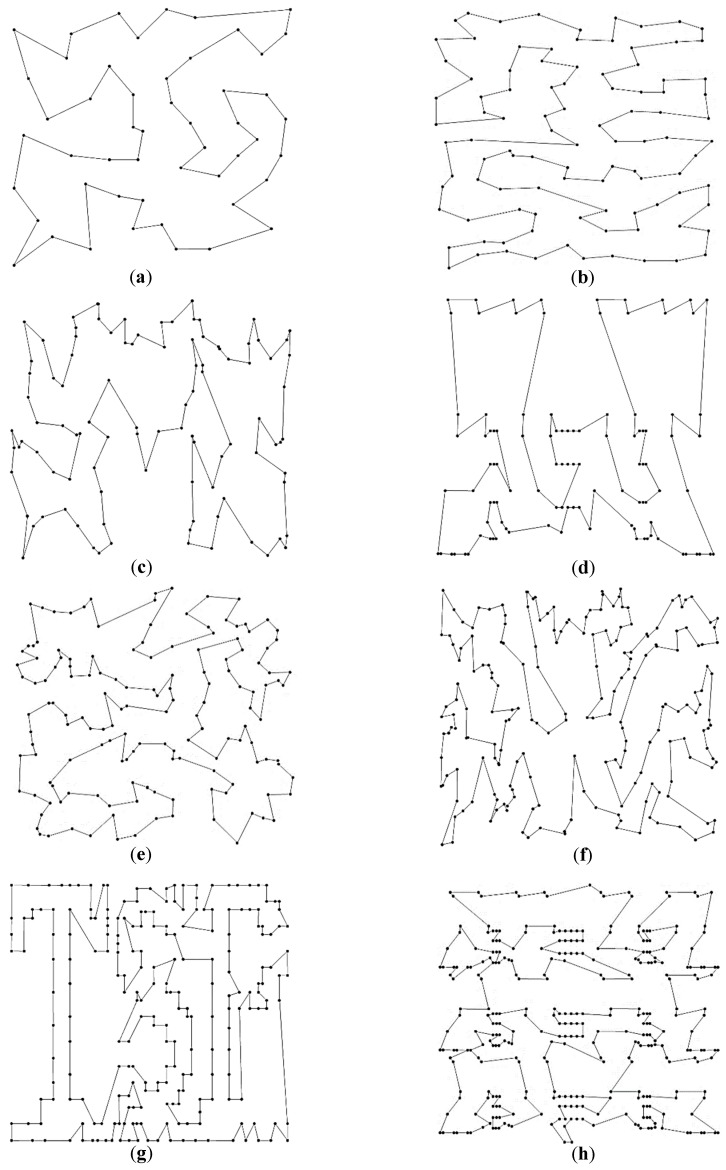
Best route for eight TSPLIB examples using the IPSO: (**a**) eil51; (**b**) rat99; (**c**) kroa100; (**d**) lin105; (**e**) ch150; (**f**) kroa200; (**g**) tsp225; (**h**) lin318.

**Figure 8 sensors-19-04620-f008:**
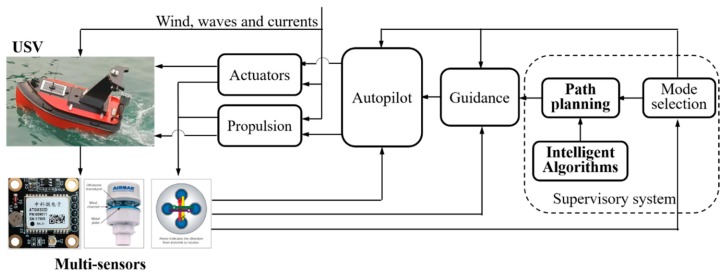
Navigation, guidance and control system of the self-developed unmanned surface vehicle (USV).

**Figure 9 sensors-19-04620-f009:**
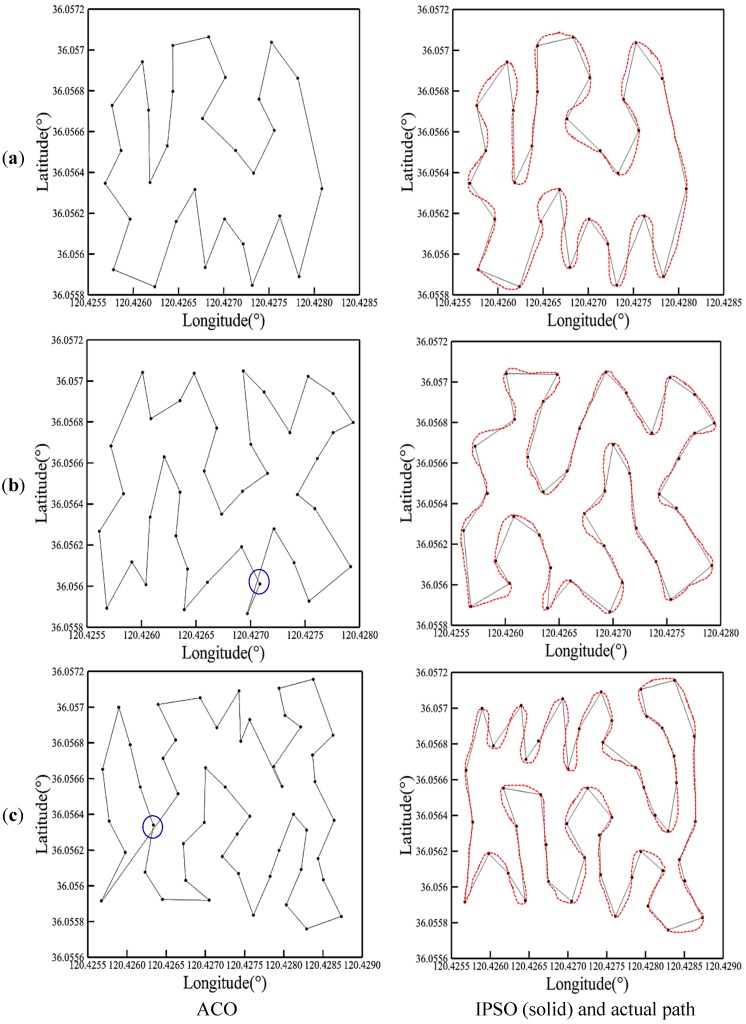
Comparison between USV trajectories planned by the ACO, IPSO algorithms, and the actual path in real sea environment: (**a**) *Q* = 30; (**b**) *Q* = 40; (**c**) *Q* = 50.

**Table 1 sensors-19-04620-t001:** Parametrization of the conventional particle swam optimization (CPSO), conventional genetic algorithm (CGA), ant colony optimization (ACO), and improved particle swam optimization (IPSO).

Algorithm	Parameters	Settings
CPSO	population size *R*	500
	personal cognition coefficient *c*_1_	constant, 2
	social cognition coefficient *c*_2_	constant, 2
	inertia weight *w*	constant, 0.9
CGA	population size *R*	500
	crossover probability *P_C_*	constant, 0.9
	mutation probability *P_M_*	constant, 0.1
ACO	population size *R*	30
	information heuristic factor *α*	constant, 1
	expectation heuristic factor *β*	constant, 5
	pheromone evaporation ratio *ρ*	constant, 0.2
IPSO	population size *R*	500
	personal cognition coefficient *c*_1_	constant, 2
	social cognition coefficient *c*_2_	constant, 2
	inertia weight *w*	constant, 0.9

**Table 2 sensors-19-04620-t002:** Computational results of CPSO, CGA, ACO and IPSO.

Methed	Problem	Eil51	Rat99	Kroa100	Lin105	Ch150	Kroa200	Tsp225	Lin318
	KOS	426	1211	21,282	14,379	6258	29,368	3916	42,029
CPSO	AVG	823.89	4153.66	85,243.19	60,624.7	30,769.4	195,276.5	25,161.71	381,854.1
	SD	45.09	171.43	4241.85	2977.68	981.38	6794.39	678.94	8776.39
	RPE (%)	93.40	242.99	300.54	321.62	391.68	564.93	542.54	808.55
	M_cri_	46	66	62	70	62	76	71	82
CGA	AVG	555.24	2442.2	48,702.81	34,357.11	18,775.58	120,657.6	16,473.08	266,979.7
	SD	32.17	166.74	3442.36	2280.54	721.97	5189.39	643.00	8550.38
	RPE (%)	30.33	101.66	128.84	138.93	200.02	310.84	320.66	535.22
	M_cri_	224	384	446	567	662	745	875	905
ACO	AVG	455.41	1339.64	23,166.06	15,253.3	6994.37	33,016.26	4300.29	48,473.74
	SD	5.92	13.75	269.04	132.12	51.75	553.96	52.99	465.02
	RPE (%)	6.91	10.62	8.85	6.08	10.17	12.42	9.81	15.33
	M_cri_	75	87	108	103	120	127	246	249
IPSO	AVG	443.68	1312.05	22,981.74	15,241.78	6836.46	31,976.55	4188.00	45,868.84
	SD	5.85	26.27	587.36	318.33	37.43	598.57	53.31	520.49
	RPE (%)	3.51	8.34	7.98	5.93	9.24	8.88	6.94	9.13
	M_cri_	46	100	113	112	135	179	263	285

* KOS is known optimal solution from TSPLIB; AVG is average route length; SD is standard deviation; RPE (%) is relatively percentage error; M_cri_ is critical iteration number.

**Table 3 sensors-19-04620-t003:** Simulation results of ACO and IPSO for three numbers of planned points.

*Q*	Algorithm	*m_cri_*	*D* (m)
30	ACO	47	1100.23
	IPSO	27	1100.23
40	ACO	59	1214.34
	IPSO	33	1200.35
50	ACO	78	1726.94
	IPSO	58	1661.23

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
