# Peer review of "Greedy Mechanism Based Particle Swarm Optimization for Path Planning Problem of an Unmanned Surface Vehicle"

_sensors, 2019, doi:10.3390/s19214620_

Round 1

Reviewer 1 Report

The paper presents an improved PSO algorithm. By introducing greedy mechanism to generate partial particles and 2-opt operation to eliminate path-crossing phenomena, the paper claims that its effectiveness in some optimization problems has been improved which is also supported with numerical examples.

In general the paper is clearly organized and the English is good. My main concern is that the paper lacks a clear theoretical explanation on the new algorithm, and the mechanism of the performance improvement is not convincing. Under which scenarios would the proposed algorithms outperform the conventional ones, as well as other recently proposed PSO algorithms (apart from the existing ones reviewed in the paper)? Would there be any downsides of the algorithms (except the slight ones mentioned in the paper)?

The comparisons with other recent PSO algorithms are important. I have noticed the authors also published some similar papers on this topic, and a careful review and study is appreciated. 

The application to USV path planning seems to be disconnected to the study of the PSO algorithm. Furthermore, path/motion planning of USVs should always take into account their motion dynamics as constraints, but this is not discussed in the paper. 

The application on USV seems to be disconnected to the study on PSO algorithm. 

Author Response

Responses to Reviewers’ Comments

We really appreciate all insightful comments and useful suggestions, which will help us to improve the quality and readability of our manuscript. The manuscript has been revised according to the comments. In order to make the revising contents more clearly, the changes according to the reviewer#1 will be highlighted in Yellow.

Reviewer #1

The paper presents an improved PSO algorithm. By introducing greedy mechanism to generate partial particles and 2-opt operation to eliminate path-crossing phenomena, the paper claims that its effectiveness in some optimization problems has been improved which is also supported with numerical examples. In general the paper is clearly organized and the English is good.

Response: Thanks for the reviewer’s positive comment. All the suggestions are fully appreciated. In our opinion, all the comments are highly constructive and useful to restructure our manuscript. We believe that the new or modified contents included in the revised version really have improved the quality of the manuscript. We hope all the modifications will fulfill the requirement to make the manuscript acceptable for publication in this archive journal.

Detailed comments:

Comment 1: My main concern is that the paper lacks a clear theoretical explanation on the new algorithm, and the mechanism of the performance improvement is not convincing.

Response: Thanks for the reviewer’s comment. We have to admit that it is really hard to derive the theoretical or mechanism analysis of intelligence algorithms in our work. Instead, we want to add more intuitive explanations on the new algorithm.

(1) It is known that the conventional PSO generates initial swarm of particles in a random way; this would result in some infeasible solutions and consequently restrict convergence speed and search efficiency. Hence, it is expected to enhance the algorithm effectiveness by integrating two local strategies, greedy mechanism and 2-opt operation, into the PSO. In our new algorithm, a greedy black box is utilized for the stage of particles initialization and particle generation at each iteration. However, since greedy mechanism only considers current circumstances for local search, it has inherent shortcomings that path crossovers are easy to arise. Hence, it is necessary to adopt 2-opt operation subsequently in order to enhance the probability of satisfactory combinations between locally optimal fragments and eliminate path crossovers.

(2) On the one hand, the utilization of greedy black box excludes a great number of infeasible solutions resulted from the randomness of traditional method and creates initial swarm of particles with relatively higher quality. On the other hand, the strategy of greedy black box makes the newly planned route consist of several locally optimal fragments with advantages of short distance, low order and high adaptability. Hence, it seems feasible and reasonable to adopt the greedy black box for particles initialization.

(3) After the update of particles’ velocity and position during an iteration, all the particles are scrambled and divided into sub groups of four. The “out of order” execution avoids repeating the later replacement to a same particle as much as possible. Meanwhile, the operation of grouping helps establish a communication platform within a subgroup or between arbitrary two subgroups. Then two particles with lower fitness values in each sub group are chosen. Their first cities are utilized by the greedy black box to generate two new particles. If the new generated particles have higher fitness values, they will replace the old ones; otherwise, the replacement will not be conducted. This operation results in the survival of the fittest to some degree and prevents the swarm of particles being generated entirely by the greedy black box, and is also called greedy selection strategy.

Please see details in Page 6, Line 197-202; Page 7, Line 220-222, 233-239.

Comment 2: Under which scenarios would the proposed algorithms outperform the conventional ones, as well as other recently proposed PSO algorithms (apart from the existing ones reviewed in the paper)? Would there be any downsides of the algorithms (except the slight ones mentioned in the paper)? The comparisons with other recent PSO algorithms are important. I have noticed the authors also published some similar papers on this topic, and a careful review and study is appreciated.

Response: Thanks for the reviewer’s comment. According to the reviewer’s comment, we try to compare our new algorithm with other recently proposed PSO algorithms. Among them, we include our last developed algorithm, namely particle swarm optimization with adaptively controlled acceleration coefficients and linearly descending inertia weight (AWIPSO) [Ref. 1]. Since it is difficult to repeat their algorithm code and reproduce their results, we only gather partial results from published papers which makes the comparison incomplete and unsystematic, as shown in the following table.

Results show that our new algorithm has a superior performance in path distance. When more planned points are considered, the difference between our optimal solution and the known optimal solution from TSPLIB has an increasing tendency. It indicates that our new algorithm still has room for improvement.

Indeed, our group is now working on a review study which presents recent PSO algorithms on path planning for autonomous vehicles. Comparative study is carried out in terms of optimal path distance, algorithm robustness and computing efficiency. We also sent emails to several experts for requesting assistance in refining comparative results. However, we have to say it consists of large amount of work and is quite time-consuming. For simplicity, we decided not to show these incomplete comparisons in this work.

In addition, the new algorithm has certain disadvantages from theoretical perspective. Once the code of departing city is determined, the particle generated by the greedy black box is unique. Furthermore, when more planned points are considered, the maximum number of iterations should be increased to guarantee the successful execution of 2-opt. Otherwise, the path crossovers could not be completely eliminated. Please see details in Page 9, Line 276-280.

Comment 3: The application to USV path planning seems to be disconnected to the study of the PSO algorithm. Furthermore, path/motion planning of USVs should always take into account their motion dynamics as constraints, but this is not discussed in the paper.

Response: Thanks for the reviewer’s comment. In the design stage, the self-developed USV is expected to be used in fields of military, scientific research and environmental protection to execute tasks, such as autonomous cruise, water quality sampling, water quality monitoring and garbage collection. It is of crucial importance to design the feasible and shortest trajectories covering all preset points for the purpose of reducing costs and enhancing efficiency. Generally, path planning can be simplified into the TSP if the prior environmental information is inaccessible and the collision-free restriction is not taken in to account. Hence, the proposed IPSO could be applied to the path planning subsystem of the USV, designing feasible trajectories to follow under real sea conditions. Meanwhile, since this work is preliminary, the factors of wind, current, waves and obstacles are not considered for simplicity. Please see details in Page 11, Line 354-359.

Reference

[1]   Xin, J.; Li, S.; Sheng, J.; Zhang, Y.; Cui, Y. Application of Improved Particle Swarm Optimization for Navigation of Unmanned Surface Vehicles. Sensors 2019, 19(14), 3096.

[2]  Arram, A.; Ayob, M. A Novel Multi-Parent Order Crossover in Genetic Algorithm for Combinatorial Optimization Problems. Computers & Industrial Engineering 2019, 133, 267-274.

[3]    Tongur, V.; Ulker, E. PSO-based Improved Multi-Flocks Migrating Birds Optimization (IMFMBO) Algorithm for Solution of Discrete Problems. Soft Computing 2018, 23(14), 5469-5484.

Table. Average values of optimal path distance of CPSO, CGA, ACO and IPSO for eight TSPLIB instances.

Methed

Problem  

Eil51

Rat99

Kroa100

Lin105

Ch150

Kroa200

Tsp225

Lin318

KOS

426

1211

21282

14379

6258

29368

3916

42029

CPSO

AVG

823.89

4153.66

85243.19

60624.7

30769.4

195276.5

25161.71

381854.1

CGA

AVG

555.24

2442.2

48702.81

34357.11

18775.58

120657.6

16473.08

266979.7

ACO

AVG

455.41

1339.64

23166.06

15253.3

6994.37

33016.26

4300.29

48473.74

AWIPSO [Ref.1]

AVG

455.91

1332.97

24524.3742

15572.82

13455.79

100975.16

5421.85

70981.44

MPOX [Ref.2]

AVG

523.27

/

33221.43

12416.63

49434.17

/

/

67034.77

MBO [Ref.3]

AVG

465.4

/

29716.13

21071.87

9460.60

48423.13

6157.57

/

IPSO

AVG

443.68

1312.05

22981.74

15241.78

6836.46

31976.55

4188.00

45868.84

Reviewer 2 Report

In this manuscript, in order to design higher-quality routes for a multi-sensor integrated USV, the authors have improved conventional particle swarm optimization algorithm by introducing greedy mechanism and 2-opt operation.  A greedy black box was established for particle initialization .Monte-Carlo  simulations of eight TSPLIB instances were conducted to compare the improved algorithm with some other existing algorithms. Computation results indicated that the improved algorithm has the best performance with the shortest route and satisfactory robustness.

The work is interesting. The contributions and novelty are sound. The detail comments are given as follows.

1: The related works should be improved. A more state-of-art survey for the related works needs to be presented. The following related works should be discussed in order to highlight the contributions of this manuscript.

[1] “Computation Rate Maximization in UAV-Enabled Wireless Powered  Mobile-Edge Computing Systems,” IEEE Journal on Selected Areas in Communications, vol. 36, no. 9, pp.1927-1941, Sept. 2018.

[2] “Resource allocation for secure UAV-assisted wireless communication systems with SWIPT,” IEEE Access, vol.7, pp. 24248-24257, 2019.

2: The contributions should be highlighted in the introduction in order make it easy to follow.

3:The table should be defined. And the figures should be improved.

Author Response

Responses to Reviewers’ Comments

We really appreciate all insightful comments and useful suggestions, which will help us to improve the quality and readability of our manuscript. The manuscript has been revised according to the comments. In order to make the revising contents more clearly, the changes according to the reviewer#1 will be highlighted in Yellow.

Reviewer #2

In this manuscript, in order to design higher-quality routes for a multi-sensor integrated USV, the authors have improved conventional particle swarm optimization algorithm by introducing greedy mechanism and 2-opt operation. A greedy black box was established for particle initialization. Monte-Carlo  simulations of eight TSPLIB instances were conducted to compare the improved algorithm with some other existing algorithms. Computation results indicated that the improved algorithm has the best performance with the shortest route and satisfactory robustness. The work is interesting. The contributions and novelty are sound.

Response: Thanks for the reviewer’s positive comment. All the suggestions are fully appreciated. In our opinion, all the comments are highly constructive and useful to restructure our manuscript. We believe that the new or modified contents included in the revised version really have improved the quality of the manuscript. We hope all the modifications will fulfill the requirement to make the manuscript acceptable for publication in this archive journal.

Detailed comments:

Comment 1: The related works should be improved. A more state-of-art survey for the related works needs to be presented. The following related works should be discussed in order to highlight the contributions of this manuscript.

[1] “Computation Rate Maximization in UAV-Enabled Wireless Powered  Mobile-Edge Computing Systems,” IEEE Journal on Selected Areas in Communications, vol. 36, no. 9, pp.1927-1941, Sept. 2018.

[2] “Resource allocation for secure UAV-assisted wireless communication systems with SWIPT,” IEEE Access, vol.7, pp. 24248-24257, 2019.

Response: Thanks for the reviewer’s comment. Since this work focuses on the path planning problem of USV, we decided not to add the above two references to the context. However, our group is now working on a review study which presents recent path planning algorithms for autonomous vehicles. We decided to cite the two valuable works in our next paper. Thanks again for the reviewer’s suggestion.

Comment 2: The contributions should be highlighted in the introduction in order make it easy to follow.

Response: Thanks for the reviewer’s comment. We have added the contributions of this work to the last but one paragraph in Section “1. Introduction”. The main contributions of this work are as follows: (1) A greedy black box is established to generate initial swarm of particles which avoids the randomness of traditional method; (2) The strategy of greedy selection guarantees particles to move towards a higher fitness level and keeps the swarm diversity; (3) 2-opt operation performs effectively in maintaining locally optimal fragments of relatively inferior particles and eliminate path crossovers; (4) The improved algorithm has been successfully applied to the path planning subsystem of a USV model with the aid of multi-sensors. Please see details in Page 2, Line 78-83.

Comment 3: The table should be defined. And the figures should be improved.

Response: Thanks for the reviewer’s comment. According to the reviewer’s comments, we have refined all the tables and figures in table layout and figure resolution. Please see corresponding modification.

Reviewer 3 Report

The paper proposes the combined algorithms to solve the path planning problems. The basic idea is to combine the conventional particle swarm optimization method with a greedy mechanism to generate partial particles and 2-opt operation to eliminate path-crossing phenomena. The greedy mechanism is first utilized for the stage of particles initialization. Then, it is adopted again along with the 2-opt operation for local search at each iteration. Monte-Carlo simulations for the well-known dataset are conducted to verify the effectiveness and reliability of the improved method in terms of solution quality, algorithm robustness, and computing efficiency.

Major comments of the paper are as follows.

1. The paper has the self-plagiarism problem with their previous works [1, 2] of the same journal. Even though there are some minor updates, some sentences of the paragraphs are the same with [1, 2].

[1] Xin, J.; Li, S.; Sheng, J.; Zhang, Y.; Cui, Y. Application of Improved Particle Swarm Optimization for Navigation of Unmanned Surface Vehicles. Sensors, 2019.

[2] Xin, J, Zhong, J, Yang, F, Cui, Y, Sheng, J, An Improved Genetic Algorithm for Path-Planning of Unmanned Surface Vehicle. Sensors, 2019.

2. Although the motivation of the combined approach is discussed in Section 2.3, it is hard to follow the main idea of the proposed scheme. Please clarify better the basic idea of the paper.

3. The mathematical notions are not well defined and written. For instance, rewrite all subscript and upper scripts of mathematical notations.  In addition, make tables for the abbreviations and the mathematical notations.

4. Some acronyms are not even well defined in the paper e.g., TSPLIB.

5. The paper requires serious proofreading due to many typos and grammatical errors.

Author Response

Responses to Reviewers’ Comments

We really appreciate all insightful comments and useful suggestions, which will help us to improve the quality and readability of our manuscript. The manuscript has been revised according to the comments. In order to make the revising contents more clearly, the changes according to the reviewer#1 will be highlighted in Yellow.

Reviewer #3

Detailed comments:

Comment 1: The paper has the self-plagiarism problem with their previous works [1, 2] of the same journal. Even though there are some minor updates, some sentences of the paragraphs are the same with [1, 2].

[1] Xin, J.; Li, S.; Sheng, J.; Zhang, Y.; Cui, Y. Application of Improved Particle Swarm Optimization for Navigation of Unmanned Surface Vehicles. Sensors, 2019.

[2] Xin, J, Zhong, J, Yang, F, Cui, Y, Sheng, J, An Improved Genetic Algorithm for Path-Planning of Unmanned Surface Vehicle. Sensors, 2019.

Response: Thanks for the reviewer’s comment. We admit that there are similar sentences in Section 4.2 with our last published papers. This similar part is a detailed description of the navigation, guidance and control system for our self-developed USV model. We apply the same USV model to test the effectiveness of our newly proposed PSO algorithm when carrying out experiments in real environment. As is known, designing and constructing another completely different USV model costs too much and is not strictly necessary. Anyway, we have updated the description of this part and added more information. Please see the modification in the modified version highlighted in Green. Please see details in Page 12, Line 366-373.

Comment 2: Although the motivation of the combined approach is discussed in Section 2.3, it is hard to follow the main idea of the proposed scheme. Please clarify better the basic idea of the paper.

Response: Thanks for the reviewer’s comment. It is known that the conventional PSO generates initial swarm of particles in a random way; this would result in some infeasible solutions and consequently restrict convergence speed and search efficiency. Hence, it is expected to enhance the algorithm effectiveness by integrating two local strategies, greedy mechanism and 2-opt operation, into the PSO. In our new algorithm, a greedy black box is utilized for the stage of particles initialization and particle generation at each iteration. However, since greedy mechanism only considers current circumstances for local search, it has inherent shortcomings that path crossovers are easy to arise. Hence, it is necessary to adopt 2-opt operation subsequently in order to enhance the probability of satisfactory combinations between locally optimal fragments and eliminate path crossovers. Please see details in Page 6, Line 197-202.

Comment 3: The mathematical notions are not well defined and written. For instance, rewrite all subscript and upper scripts of mathematical notations.  In addition, make tables for the abbreviations and the mathematical notations.

Response: Thanks for the reviewer’s comment. We have carefully checked the manuscript and made corrections according to the reviewer’s comment. Please see the corresponding modifications.

Comment 4: Some acronyms are not even well defined in the paper e.g., TSPLIB.

Response: Thanks for the reviewer’s comment. We have carefully checked the manuscript and made corrections according to the reviewer’s comment. Please see details in Page 21, Line 494.

Comment 5: The paper requires serious proofreading due to many typos and grammatical errors.

Response: Thanks for the reviewer’s comment. Following the reviewer’s suggestion, we carefully checked the manuscript entirely ourselves to eliminate all the simple grammar errors such as the wrong usage of the verb conjugations and other sentences which are not easy to be understood. All the responses in this file are also carefully checked for smoothness. Please checked all the highlighted changings in the revised manuscript.

Round 2

Reviewer 1 Report

I thank the authors for revising the paper, and some of my comments are addressed in this version. However, I still have some further comments on this paper. 

The authors have detailed some disadvantages (from theoretical perspective) of the proposed algorithm (see Lines 276--280); however there are no discussions or remedy on how such disadvantages could be relieved or avoided. Some discussions on improving the algorithm that avoid disadvantages will be appreciated.  The authors have some sequential papers on the same topics published recently in the same journal (this is also pointed out by other reviewers). It will be helpful to further clarify their connections, new contributions, and avoiding any possible overlap from these research papers. Otherwise, it seems repeating the same problem via some modifications of PSO or other intelligence algorithms, applied to the same problem. 

Reviewer 3 Report

I do not find any detailed responses to each concern of the reviewer. Furthermore, the self-plagiarism problem is not yet resolved. In Section 1, 2.2, 3.1, 4.2, I still find the number of same sentences with respect to their previous papers [1, 2]. 

[1] Xin, J.; Li, S.; Sheng, J.; Zhang, Y.; Cui, Y. Application of Improved Particle Swarm Optimization for Navigation of Unmanned Surface Vehicles. Sensors 2019.

[2] Xin, J, Zhong, J, Yang, F, Cui, Y, Sheng, J, An Improved Genetic Algorithm for Path-Planning of Unmanned Surface Vehicle. Sensors, 2019.

Round 3

Reviewer 1 Report

I thank the authors for revising the paper. My two comments in the last round of review have been addressed and I do not have further comments. 

I recommend its publication if space is available. 

A small suggestion: the authors may wish to revise some overlapping contents to avoid repeating the same paragraphs as in their previous papers. 

Reviewer 3 Report

The authors revise the paper based on the comments.